# Small Molecule Inhibitors in the Treatment of Rheumatoid Arthritis and Beyond: Latest Updates and Potential Strategy for Fighting COVID-19

**DOI:** 10.3390/cells9081876

**Published:** 2020-08-11

**Authors:** Magdalena Massalska, Wlodzimierz Maslinski, Marzena Ciechomska

**Affiliations:** Department of Pathophysiology and Immunology, National Institute of Geriatrics, Rheumatology and Rehabilitation, 02-637 Warsaw, Poland; magdalena.massalska@spartanska.pl (M.M.); wlodzimierz.maslinski@spartanska.pl (W.M.)

**Keywords:** small molecule inhibitors, tsDMARDs, COVID-19, rheumatoid arthritis, clinical trials, JAK inhibitors, side effects

## Abstract

The development of biological disease-modifying antirheumatic drugs (bDMARDs) and target synthetic DMARDs (tsDMARDs), also known as small molecule inhibitors, represent a breakthrough in rheumatoid arthritis (RA) treatment. The tsDMARDs are a large family of small molecules targeting mostly the several types of kinases, which are essential in downstream signaling of pro-inflammatory molecules. This review highlights current challenges associated with the treatment of RA using small molecule inhibitors targeting intracellular JAKs/MAPKs/NF-κB/SYK-BTK signaling pathways. Indeed, we have provided the latest update on development of small molecule inhibitors, their clinical efficacy and safety as a strategy for RA treatment. On the other hand, we have highlighted the risk and adverse effects of tsDMARDs administration including, among others, infections and thromboembolism. Therefore, performance of blood tests or viral infection screening should be recommended before the tsDMARDs administration. Interestingly, recent events of SARS-CoV-2 outbreak have demonstrated the potential use of small molecule inhibitors not only in RA treatment, but also in fighting COVID-19 via blocking the viral entry, preventing of hyperimmune activation and reducing cytokine storm. Thus, small molecule inhibitors, targeting wide range of pro-inflammatory singling pathways, may find wider implications not only for the management of RA but also in the controlling of COVID-19.

## 1. Introduction 

Rheumatoid arthritis (RA) is an aggressive immune-mediated disease with a worldwide prevalence of approximately 0.5–1% of population. RA is more common in women and may occur at any age, with peak incidence occurring at 50–60 years of age [1]. RA is characterized by joint involvement, high morbidity, progressive disability and increased mortality. In fact, more than 30% of RA patients become work disabled after 10 years. In addition, RA patients have reduced life expectancy on average by 5–10 years [2,3]. The estimated total costs of RA treatment are around 45 billion euro in Europe and 41 billion euro in the United States [4]. Therefore, RA results in a significant burden for patients and healthcare systems. The underlying cause for RA is still unknown and current therapies are more or less effective in controlling symptoms, but still fail to cure the disease. 

The development of biological disease-modifying antirheumatic drugs (bDMARDs) represent a breakthrough in RA treatment. Indeed, bDMARDs were pioneered by TNF inhibitor (TNFi), which was first approved by Food and Drugs Administration (FDA) in 1998. Subsequently, other biologic agents blocking pro-inflammatory cytokines including IL-1, IL-6 or neutralizing antigens on the cell surface such as CTLA-4 or CD20 were developed. Importantly, in 2012 FDA approved new class of targeted synthetic DMARDs (tsDMARDs), which were developed to target a particular molecular structure. This class of drugs is also known as small molecule inhibitors. The first FDA approved tsDMARD was tofacitinib inhibiting Janus kinases (JAKs). This new class of drugs represent a large family of small molecules targeting the several types of kinases including JAK, a mitogen-activated protein kinase (MAPK), spleen tyrosine kinase (SYK)-Bruton’s tyrosine kinase (BTK) (SYK-BTK) or nuclear factor, such as NF-κB. Recent American College of Rheumatology (ACR) and the European League Against Rheumatism (EULAR) guidelines have recommend the use of bDMARDs and tsDMARDs to treat patients with moderate-to-severe disease activity [5]. Unfortunately, these treat-to-target medicines are usually recommended only for patients that do not respond to methotrexate or other conventional synthetic DMARDs (csDMARDs), due to high cost of treatment. For instance, the annual cost of upadacitinib (JAK inhibitor—JAKi) treatment reaches $59,860 [6]. Of note, all three FDA approved inhibitors of JAKs for RA (tofacitinib, baricitinib and upadacitinib) are administered orally, which may be preferable to patients compared to biologic agents which are administered intravenously or via subcutaneous injection. Small molecules inhibitors have shorter half-life than bDMARDs and need to be taken either once or twice a day. In general, small molecules inhibitors are also smaller (≤500 Da size) compared to biologics (>1000 Da size). On the other hand, tsDMARDs offer wide protection against pro-inflammatory cytokines, in opposite to bDMARDs which block specific extracellular molecules.

Importantly, recent event of severe acute respiratory syndrome coronavirus 2 (SARS-CoV-2) outbreak suggests that some tsDMARDs, targeting JAK1 and bDMARDs targeting IL-1, IL-6, GM-CSF are being investigated as potential therapies for coronavirus disease 2019 (COVID-19) in order to block cytokine storm. On the other hand, RA patients with an immunocompresed system might be at higher risk for developing severe form of COVID-19; thus, administration of tsDMARDs should be rigorously monitored.

This review provided the latest update on development of tsDMARDs, their efficacy and safety as a strategy for RA treatment. On the other hand, we have highlighted the risk and adverse events (AEs) of tsDMARDs administration including infections and thromboembolism. Finally, we discussed the latest guidelines and recommendations of tsDMARDs treatment in the light of pandemic event of SARS-CoV-2 and subsequent COVID-19 illness.

## 2. tsDMARDs Based on JAKs/MAPKs/NF-κB/SYK-BTK-Targeted Therapy 

### 2.1. Signaling of JAKs/MAPKs/NF-κB/SYK-BTK

Cellular exposure to cytokines, chemokines, growth factors, pathogen-associated molecular patterns (PAMPs) or antigens results in receptor ligation on the cell surface [7]. Signaling cascades that are subsequently initiated lead to altered expression of genes involved in inflammation and other cellular processes mounting an adequate response to stimuli [7]. These pathways comprise of JAKs together with Signal Transducers and Activators of Transcription (STAT) pathway, MAPKs pathway, NF-κB pathway and SYK-BTK-signaling [7] (Figure 1). Deregulated JAK/STAT activation is now accepted as playing a critical role in perpetuating pathology of RA through its capacity to up-regulate gene expression of proinflammatory cytokines [8]. Importantly, JAK/STAT pathway activation can also result in activation of the MAPK pathway (by “cross-talk”), which is partially responsible for neoangiogenesis and up-regulation of matrix metalloproteinases (MMP), that together with other enzymes is responsible for degradation of articular cartilage extracellular matrix proteins. Unregulated development of osteoclastogenesis results in destruction of subchondral bone via proteinase-mediated degradation [8]. Finally, hyperactivation of NF-κB and overstimulation of the Phosphoinositide 3-kinases (PI3K)/Akt/Protein Kinase B/mammalian target of rapamycin (PI3K/Akt/PKB/mTOR) “cell survival” pathway are probably responsible for the apoptosis resistance, characteristic for RA inflamed synovial tissue [8]. Therefore, molecules building these signaling pathways become attractive targets for the development of new therapies based on tsDMARDs.

#### 2.1.1. Signaling of JAKs

The JAK family, together with the transcription factors of STAT, compose signal transmission pathway between extracellular receptors (receptors for cytokines) and cell nucleus. The JAK-STAT pathway is activated by more than 50 different cytokine receptors and results in transcriptional regulation of genes that coordinate cell proliferation, differentiation, activation and metabolic homeostasis. It is the common pathway for intracellular signal transduction of pro-inflammatory mediators like cytokines (IL-6, IL-11, IL-12, IL-23), type I and II IFNs, all γ-chain cytokines (IL-2, IL-4, IL-7, IL-9, IL-15, IL-21), hematopoietic growth factors as IL-3, IL-5 and GM-CSF, erythropoietin and thrombopoietin [9]. There are four different types of human JAKs (JAK1, JAK2, JAK3 and TYK2) and seven inactive cytoplasmic proteins STAT (STAT1, STAT2, STAT3, STAT4, STAT5a and b and STAT6) that compose JAK/STAT pathway [10]. Activation process of signaling cascade results in accumulation of JAKs and STAT molecules into homo-, hetero-, or multimers and the final, transcriptional effects is dependent on the type of JAK and STAT involved in building specific multimer. Finally, phosphorylated STAT dimers translocate to the nucleus where they act as a transcription factors and regulate process of transcription of targeted genes. Cytokines are directly implicated in each phase of RA pathogenesis—by promoting autoimmunity, by maintaining chronic inflammatory synovitis and finally driving the destruction of the adjacent joint tissue [11]. Additionally, elevated expression of STAT3, STAT1, STAT4, STAT6 and JAK3 detected in RA, suggests that their activation stimulate pathogenesis of RA [12,13,14,15]. Thus, molecules that regulate the JAK-STAT pathways are proposed to be real therapeutic drugs for the treatment of RA.

#### 2.1.2. Signaling of MAPKs

There are three subfamilies of MAPKs: extracellular signal-regulated kinases (ERK), c-Jun amino-terminal kinases (JNK), and p38 kinase (p38) [16]. The MAPK pathway begins with MAPK kinase kinases (MAP3K), which phosphorylate and activate the MAPK kinases (MKK). The MKK then phosphorylate MAPK, which subsequently activate various transcription factors. This pathway regulates fundamental cellular processes like cell cycle regulation, apoptosis, cell aging and the production of cytokines such as IL-10. The p38 is a key protein in the regulation of the pro-inflammatory response and thus was one of the first protein kinase investigated as a therapeutic target in autoimmunity and inflammation [16]. Activated ERK, JNK and p38 are present in synovial tissue of RA patients, suggesting that they play an important role in this autoimmune disease [17,18]. Additionally, TNF, IL-1β and COX-2 are among the most important pro-inflammatory mediators regulated by p38, which inhibition has been demonstrated to result in clinical benefit in RA patients [19]. The biological processes regulated by p38 kinase suggest a wide variety of potential indications for inhibitors, but level of complexity has proven challenging to the drug discovery effort [19].

#### 2.1.3. Signaling of NF-κB

NF-κB represents a family of inducible transcription factors (p50, p52, p65, RelB and c-Rel) regulating many genes involved in processes of immune and inflammatory responses. The NF-κB proteins are normally sequestered in the cytoplasm by inhibitory proteins, including IκB. NF-κB target inflammation not only directly by triggering the production of chemokines, adhesion molecules and pro-inflammatory cytokines, but also by modulating cell proliferation, survival, morphogenesis and differentiation [20]. There are two signaling pathways leading to activation of NF-κB: canonical and alternative. The canonical pathway responds to diverse stimuli, including ligands of cytokine receptors, pattern recognition receptors (PRRs), TNF receptor superfamily (TNFRSF), T-cell receptor (TCR) and B-cell receptor (BCR). Alternative pathway selectively responds to a specific group of stimuli including ligands for a subset of TNFRSF (RANK, CD40 and BAFFR) [20]. Activated NF-κB in synovial tissue of RA patients have been described many years ago [21]. NF-κB contributes to RA pathogenesis by acting in many different cell types. First, it mediates the induction of pro-inflammatory cytokines, such as TNF, IL-1 and IL-6 in monocytes and macrophages. Many of these cytokines are able to activate NF-κB in innate immune cells and fibroblasts leading to further dissemination of inflammation [20]. Second, NF-κB promotes Th17 differentiation and support survival of self-reactive B cells both cell populations being strongly involved in RA development [22,23].

#### 2.1.4. Signaling of SYK and BTK

SYK and BTK are cytoplasmic non-receptor tyrosine kinases, transmitting signals from a different cell surface receptors like BCR, Fc receptors, CD74 and integrins [24]. Receptor crosslinking followed by cascade of enzymatic activation leads to cooperation of recruited SYK with BTK to activate phospholipase C-gamma2 (PLC-γ2), which finally results in MAPK and Phosphoinositide 3-kinase (PI3K) dependent downstream signaling cascades regulating diverse biological processes like cell growth, proliferation, differentiation and cytoskeletal remodeling [24]. The SYK family comprises two members: zeta-chain-associated protein kinase 70 (ZAP70) and SYK kinase. ZAP 70 expression is limited to T lymphocytes and NK cells, while SYK is expressed in hematopoietic cells, mast cells and synoviocytes [16]. SYK is activated in synoviocytes by pro-inflammatory cytokines like TNF and IL-1, which induce JNK activation and IL-6 production and results in IL-12 and IL-13 synthesis, stimulation of proliferation, differentiation, survival, degranulation and phagocytosis [25]. The BTK family has four members and is expressed in all hematopoietic cells and lymphocytes except for T cells and mature plasma B cells and is basic for lymphopoiesis [24]. Phosphorylated SYK was detected in peripheral blood B cells and synovial tissue of RA patients [26,27]. B cells and autoantibodies produced by most RA patients, mainly anti-citrullinated protein/peptide antibody (ACPA) and rheumatoid factor (RF), play a pivotal role in the pathogenesis of RA. As SYK functions as a key molecule in B cell receptor signaling, while BTK is fundamental for regulation of B cell proliferation and activation process; thus, both kinases were proposed as therapeutic target in RA treatment [28].

### 2.2. Clinical Studies of JAKs/MAPKs/NF-κB/SYK-BTK Inhibitors

#### 2.2.1. JAKs Inhibitors

First-generation JAKi affected a broad spectrum of signaling pathways of cytokines (pan-inhibitors), whereas second-generation JAK inhibitors aim to target selectively the chosen pathway, which limit the activity of much smaller subset of cytokines and maintain the signaling via other not-inhibited JAK-dependent pathways [29]. The main reason for selective JAKi development was the incidence of AEs, observed during pan-inhibitors treatment [29,30]. There are currently a series of JAKi in development for inflammatory indications and three pan-inhibitors had been approved for the treatment of RA: tofacitinib, baricitinib and peficitinib [30].

Tofacitinib, targeting JAK1 and JAK3, and JAK2 to a lesser extent, was the first JAKi approved by FDA and The European Medicines Agency (EMA) for treatment patients with moderate to severe RA, failing initial treatment with methotrexate (MTX) or other csDMARDs. It received its first regulatory approval for the treatment by FDA in 2012 under the trade name Xeljanz [30]. It improved disease activity in patients with RA who were receiving MTX or other non-biologic DMARDs [31,32]. In monotherapy, tofacitinib was superior not only to placebo, but also to MTX, in reducing signs and symptoms of RA [33,34]. Tofacitinib with MTX had a clinically meaningful improvements of RA signs in patients refractory to TNFi [35]. In head-to-head trial (ORAL Strategy) tofacitinib and MTX combination therapy was non-inferior to adalimumab (TNFi) and MTX in patients with an inadequate response to MTX, but clinical and radiographic treatment effects were sustained in patients receiving tofacitinib and MTX till 24 months [36,37]. 

Baricitinib is a potent, reversible and selective JAK1/JAK2 inhibitor with 100-fold higher selectivity for JAK1/JAK2 over JAK3, approved by FDA and EMA under the trade name Olumiant in 2018 [30]. It demonstrated beneficial treatment, as compared with placebo and adalimumab in patients refractory for MTX and different bDMARDs therapy [38,39]. In other studies, baricitinib appeared to be very effective in monotherapy and not inferior to baricitinib and MTX combine therapy [40,41]. RA-BEAM study showed that baricitinib provided enhanced improvement in pain and physical function in patients with well-controlled RA, suggesting it may produce effects beyond immunomodulation [42].

Peficitinib is the latest JAKs pan-inhibitor developed for the inflammatory indications, which received its regulatory approval for the RA treatment by the Pharmaceuticals and Medical Devices Agency (PMDA) in 2019, under the trade name Smyraf [30]. Peficitinib is inhibitor of JAK3 enzymatic activity and JAK1/3-mediated cell proliferation [43]. Its selectivity for JAK family kinases is similar to that of tofacitinib, but slightly less potent for JAK2 [43]. This drug appeared to be effective and safe in monotherapy or in combination with csDMARDs in patients with moderate to severe RA [44,45]. Peficitinib showed significantly improved efficacy compared with placebo and an acceptable safety profile in Asian patients with RA who had an inadequate response to csDMARDs or bDMARDs (RAJ3 study) or MTX (RAJ4 study) [46,47,48]. Recently, published results of long-term administration of peficitinib suggested that it may be an effective and safe long-term treatment option for Asian patients with RA [49]. Peficitinib was approved for the management of RA in Japan in 2019 [50].

Ruxolitinib (INCB018424) is a potent and selective JAK1/JAK2 inhibitor, with 130-fold selectivity higher for JAK1/JAK2 over JAK3. The drug was approved by FDA for the treatment of patients with myelofibrosis under the trade mark Jakafi in 2011 [30]. The safety, tolerability and efficacy of ruxolitinib was evaluated in RA patients in phase II clinical trial (NCT00550043). 

Selective JAK inhibitors have been recently reported as a result of increasing knowledge of the importance of kinase selectivity for safety, along with its role in disease course [30]. For example, JAK2 functions as a homodimer to play very important role in red blood cell formation. That is why side effects including anemia, neutropenia and thrombopenia were observed during inhibition of JAK2 within treatment with pan-inhibitors of the JAKs enzyme family. 

Upadacitinib (ABT-494) was developed as an efficient JAK1 inhibitor for the treatment of moderate to severe RA [30]. It is a first selective JAKi approved by the FDA and EMA for RA treatment (since mid-2019) [6]. Upadacitinib was reported to improve RA signs in patients with an inadequate response to MTX (BALANCE 2) or TNFi (BALANCE 1) [51,52]. It led to significant improvement of clinical signs and symptoms in patients with inadequate response to csDMARDs, including MTX, sulfasalazine or leflunomide (SELECT-NEXT, SELECT-SUNRISE), as well as bDMARDs (SELECT-BEYOND) [53,54,55]. Statistically significant improvements in clinical and functional outcomes were also showed in upadacitinib monotherapy in RA patients with inadequate response to MTX (SELECT-MONOTHERAPY), although existing recommendations for management of RA do not include any novel DMARD monotherapy as a part of the strategy [56]. Further investigation, SELECT-COMPARE trial, support the results of upadacitinib superior to placebo and adalimumab for RA signs including radiographic progression in RA patients with inadequate response to MTX [57,58]. Overall, safety profile of upadacitinib was generally similar to adalimumab [59].

The selectivity of another JAK1 inhibitor, filgotinib (GLPG0634), is 30-fold higher for JAK1 over JAK2 [60]. Filgotinib appeared to be effective and safe in patients with insufficient response to MTX, as investigated in two double blind, placebo controlled phase IIa trials [61]. It was also effective in monotherapy (DARWIN 2) and in combination with MTX (DARWIN 1), as well as in patients with active RA who had an inadequate response or intolerance to one or more bDMARDs (FINCH2) [62,63,64]. Filgotinib treatment decreased multiple biomarkers, which have a key role in immune response (IL-6, IFN-γ, TNF, IL-12, IL-17A, IL-1β), matrix degradation (MMP1, MMP3), angiogenesis (VEGF), recruitment and adhesion of leukocytes (CXCL10, CXCL13) [65]. Administration of filgotinib did not modulate the subsets of NK and T cells, but slightly increased B cell number, which is not fully understood [65].

There have been many selective JAKi developed recently and a few have been investigated in RA patients. Itacitinib was developed as selective inhibitor of JAK1, with more than 20-fold selectivity for JAK1 over JAK2 [30]. The safety, tolerability and efficacy of itacitinib was evaluated in phase II study (NCT01626573) in RA patients, but results have not been posted. Ritlecitinib (PF-06651600) is an irreversible inhibitor of JAK3 and the tyrosine kinase expressed in hepatocellular carcinoma (TEC) kinase family [66]. The efficacy and safety of ritlecitinib were evaluated in RA patients, who are seropositive for ACPA and/or RF with inadequate response to MTX. Ritlecitinib was generally well-tolerated and its treatment was associated with significant improvements in RA disease activity [66]. Decernotinib (VX-509) is a potent inhibitor of JAK3 developed by Vertex Pharmaceuticals. Four clinical studies evaluating efficacy of VX-509 in patients with RA had been completed, including phase II/III open label extension study (NCT01830985), investigating long-term safety and efficacy in subjects with RA, but no results were posted.

#### 2.2.2. MAPKs Inhibitors

MAPK inhibitors, tested in clinical trials in RA patients, were mainly based on p38 kinase inhibition and were believed to be the ideal target for oral therapy. The p38 MAPK inhibitors, including VX-702, SCIO-469 and ARRY-371797, have mostly failed in clinical trials due to lack of efficacy and potential AEs [67]. The phase II clinical trials of PH-797804, dilmapimod (SB-681323), BMS-582949 and p38 inhibitor developed by Hoffmann–La Roche has finished, but safety and efficacy data in RA treatment have not been published. Pamapimod was another p38α inhibitor, which, despite promising preclinical data, yielded disappointing results in clinical studies [68,69]. MAPK inhibitors have not succeeded in clinical trials due to the pleiotropic effects on the immune system stemming from p38α MAPK inhibition, which was not predicted by preclinical murine inflammation studies [70].

#### 2.2.3. NF-κB Inhibitors

Iguratimod (T-614) which inhibits activation of NF-κB or RelA (p65), is a novel tsDMARD, approved for RA treatment only in Japan and China [71]. Although iguratimod is recommended by the Asia Pacific League of Associations for Rheumatology (APLAR) in the treatment guidelines, the actual targets of the drug are still unknown. Iguratimod significantly inhibited the initiation and progression of RA by multiple mechanisms including regulation of T cell subsets differentiation, inhibition of human antibody secreting cells and inhibition of bone resorption [72,73,74,75]. The results of clinical studies have reported the safety and efficacy of iguratimod in monotherapy and in combination therapy with MTX [76,77].

#### 2.2.4. SYK-BTK Inhibitors

The first SYK inhibitor investigated in RA patients, fostamatinib, although promising in murine models of RA, failed in phase III trials [70]. However, several BTK inhibitors approved for cancer treatment have been repurposing for the treatment of RA and entered clinical studies (Table 1). Evobrutinib is a novel and highly selective irreversible BTK inhibitor targeting BCR and Fc receptor-mediated signaling and consequently inhibiting activation of human B cells, monocytes and basophils [78,79]. In mouse model of RA evobrutinib administration resulted in reduction of disease severity and histological damage, although did not reduce autoantibodies [78]. Safety and efficacy of evobrutinib in subjects with RA with stable MTX therapy or with inadequate response to MTX was evaluated in a phase IIa-b, randomized and double-blind studies (NCT02784106, NCT03233230). Tirabrutinib (GS-4059/ONO-4059) is a selective oral BTK inhibitor with clinical activity against many relapsed/refractory B-cell malignancies [80]. Safety and pharmacokinetics of GS-4059 in healthy volunteers and RA patients were evaluated in phase I placebo-controlled randomized study (NCT02626026), but no results have been posted. Spebrutinib (CC-292), although significantly reducing markers of chemotaxis and osteoclast activity, did not reach the statistically significant ACR 20 criteria (ACR20) response rate at week 4 in RA patients treatment [81]. HM71224 is a potent small molecule inhibitor BTK. In the first-in-man (FIM) study of the compound (NCT01765478), safety and tolerability of HM1224 were investigated, but no results were posted.

Small molecule therapy is relatively new treatment so their position in therapeutic hierarchy is not strongly placed. In RA treatment, the most recent EULAR recommendations from 2016 [82] placed JAKi in the second line, after failure of a first line csDMARDs (particularly MTX) and presence of unfavorable prognostic factors [83]. The ACR recommendations from 2015, which are more strict, assaulted the use of JAKi as the third choice treatment, in case of failure of csDMARDs or TNFi plus MTX or non-TNFi biologic (abatacept, rituximab, tocilizumab) plus MTX [5]. The reason why JAKi therapy is placed behind MTX or bDMARDs is lack of enough long-term tolerance data. However, head-to-head trials between JAKi and TNFi did not reveal clinically important differences in efficacy [84]. Because of insufficient data on oral kinase inhibitors, recent EULAR recommendations are to avoid usage of those drugs in pregnancy [85].

### 2.3. Side Effects of JAKs/MAPKs/NF-κB/SYK-BTK Inhibitors

Most AEs noted during JAKi treatment were expected and could be explained by the known mechanisms of action of cytokines, inhibited during the intervention, but there are still some AEs that are more difficult to be clarified [10]. Especially, treatment with first-generation JAKi, targeting a wide spectrum of cytokines (potentially 57 cytokines), resulted with AEs. 

JAKi are immunosuppressive treatment and thus the infectious side effects are the most expected. Pivotal safety studies of tofacitinib and baricitinib showed that the most common infections observed were those of upper airways, followed by herpes zoster infection [86,87,88,89,90]. Herpes zoster infection risk was higher than observed with bDMARDs and infections appeared at twice the rate seen in patients on biologics [10,91]. Increased zoster risk with JAKi may result from inhibition of type I IFN, which signal through JAK1. Dose-dependent increases in zoster risk without an increased risk for serious infections have been observed in patients with systemic lupus erythematosus with type I anti-IFN antibodies. This suggests an on-target mechanism, rather than generalized immune suppression [89,92]. Knowing antiviral role of IFNs, increased risk of herpes zoster is not surprising during JAKi treatment; however, it is not clear why this particular viral infection is increased [10]. Thus, it is advisable to vaccinate patients against herpes zoster before starting therapy with JAKi [10]. Rates of short-term serious AEs (within 6 months) were generally comparable across all treatments, including JAKi, adalimumab, and csDMARDs. Infections (e.g., upper respiratory tract infection, bronchitis, nasopharyngitis, urinary tract infection) were the most common AEs during treatment. Based on long-term (1 year or more) trial data, upadacitinib, tofacitinib, and baricitinib showed comparable overall safety profiles [6].

There is increased risk of deep venous thrombosis (DVT) and pulmonary embolism (PE) in patients with RA. Importantly, cardiovascular (CV) events including DVT and PE are leading cause of death in RA patients [10,93]. RA patients have a 69% higher risk of CV disease, including stroke and myocardial infarction, and a 60–140% higher risk of venous thrombo-embolism (VTE), suggesting that persistent inflammation in RA can significantly contribute to increased CV risk [94,95]. Thus, management of inflammation by all DMARDs has led to significant improvement in the clinical CV outcome of patients with RA [93]. JAK2 inhibition obviously perturbs thrombopoietin signaling and platelet homeostasis, but its relationship to thrombosis now is unclear [10]. Initially the incidence of Major Adverse Cardiovascular Events (MACE) observed in both baricitinib and tofacitinib pooled safety population was low and stable [86,89]. Increased risk of thromboembolic events was observed; however, this is only with the highest dosage of each drug: 4mg for baricitinib [96] and 10mg for tofacitinib [86]. In 2018, the FDA Adverse Event Reporting System (FAERS) assessed postmarketing reporting rates for related thromboembolic risk in tofacitinib and ruxolitinib [97]. FAERS data indicated that pulmonary thrombosis, but neither DVT nor PE may potentially be a class-wide issue for JAK inhibitors [97]. Additionally, portal vein thrombosis may be a potential risk for ruxolitinib [97]. Following recent postmarketing safety trials triggering concerns of blood clots in the lungs and death in RA patients treated with high dose of tofacitinib, both FDA and EMA issued new boxed warnings for tofacitinib 10 mg twice-daily doses [98,99,100]. Their recommendations are to avoid such use in patients with a higher risk of thrombosis (older age, obesity, medical history of DVT/PE, or immobilization after surgery). The very recently published study presents data on the suspected adverse drug reactions (SADRs) resulted from the analysis of individual case safety reports (ICSRs) for tofacitinib and baricitinib retrieved from the World Health Organisation (WHO) global database: VigiBase [101]. For tofacitinib, the majority of reports came from the US (79.6%), followed by Canada (11.9%) and Europe (3.3%), while 97.2% of ICSRs for baricitinib were from Europe and none from US or Canada [97]. This real-world study identified that patients with reported DVT or pulmonary thrombosis (PT) or PE generally had risk factors associated with thromboembolic events (older age, reporting of contraceptives, antidepressants, antithrombotic agents) [101]. Although those patients may have developed DVT or PT/PE independently of tofacitinib, the drug treatment might be an additive risk factor. While in Europe, tofacitinib treatment was associated with an elevated reporting of DVT and PE, increased reporting of PT was observed in US. Similar elevated reporting for baricitinib was observed in Europe. These results support the current recommendations for cautious use of tofacitinib in patients with high thromboembolic risk and suggest the re-examination of the use of baricitinib 4 mg in Europe [101]. The caution dealing with baricitinib used in 4mg is supported by FDA decision to limit approval only to the 2 mg formulation and results from the resent meta-analysis, which suggest higher occurrence of thromboembolic events with a 4 mg dose of baricitinib than a 2 mg dose [102]. The real-life data from the US CORRONA registry published in 2019 demonstrated that the risk of VTE in patients receiving tofacitinib versus those treated by TNFi was not statistically significant, although numerically higher [103]. Thromboembolic safety of JAKi requires further ongoing real-world assessment to determine if a class- and dose-relationship exist, especially in the context of increased clinical usage of JAKi and recent FDA- and EMA-approved upadacitinib in August 2019 [101].

Most JAKi are likely to induce cytopenias, decreased neutrophil counts and anemia because of their more or less specific inhibition of JAK2 and erythropoietin together with other hematopoietic growth factors as IL-6 and IL-11 signaling [38,43,45,51,63,83,86,89]. However, ritlecitinib, a covalent JAK3 inhibitor with high selectivity over the other JAK isoforms JAK1 and JAK2, does not inhibit JAK2 [66]. The fact that ritlecitinib spares inhibition of JAK2 cytokine signaling, connected with some hematologic AE, has made this selective inhibitor an attractive therapeutic intervention. Development of highly selective JAKi, not inhibiting JAK2 indeed might be a key to avoid thromboembolism, one of the most dangerous AE. Interestingly, thrombocytopenia observed during JAKi treatment does not seem to result from JAK2 inhibition in platelets and megakaryocytes but rather in progenitor cells [104]. However, despite the fact that neutrophils dropped after tofacitinib or baricitinib treatment, this has not been correlated with an increased infection risk [89]. Anemia observed after upadacitinib treatment (selective JAK1 inhibitor) may suggest that this drug also inhibits JAK2, particularly when used in high dosage (IC_50_ = 0.043 μM for JAK1 vs. IC_50_ = 0.2 μM for JAK2) [29,105]. 

RA patients generally have an increased incidence of cancers, including lymphoma [10]. Oncological and cardiovascular risk resulting from JAKi treatment need long term observation but the current data are reassuring. At present, the malignancy risk appears to be similar to those reported in other RA therapeutic treatment (etanercept, tocilizumab, adalimumab) and remained stable over time [89]. However, as IFNs and IL-12 take part in controlling tumor development, long blockade of JAK/STAT signaling might support malignancy development [106].

JAKs signaling participate in metabolism regulation [10]. Thus, the use of JAKi can be associated with increased liver enzymes and gastrointestinal perforation observed during JAKi treatment of RA patients, but this concept has not been yet firmly established [107]. JAKi treatment increase in total cholesterol, low-density lipoprotein (LDL) and high-density lipoprotein (HDL) particles [108]. There is observed changed lipid profile in RA patients resulting from the disease (increased and driven probably by IL-6 catabolism of LDL), but JAKi seem to restore it [109]. Despite the increased level of LDL, HDL and total cholesterol in RA patients, LDL/HDL ratio stayed stable or decreased in all JAKi studies, suggesting small impact on long term cardiovascular risk. It seems possible that JAKi can limit vascular damage by decreasing inflammation despite increasing cholesterol levels [110]. Additionally, JAKi targeting JAK2 (mainly baricitinib, but to some extend also tofacitinib) were shown to increase weight. It can be explained by the fact that JAK2 is an element of intracellular IGF-1/GH axis signaling weight gain [83,86,89].

AEs observed in RA patients taking pamapimod (MAPKs inhibitor), including infections, skin disorders, elevated levels of liver enzymes and gastrointestinal disorders, resulted in drug withdrawal [69]. Although the C-reactive protein (CRP) level was initially decreased during the treatment with MAPKs inhibitors, the effect was transient and return to baseline level [67,69,111]. This suggests a strong biologic adaptation, that allows the escape from this pathway and made p38 MAPK inhibition not useful for the treatment of RA [111].

The most commonly reported AEs during iguratimod (NF-κB inhibitor) therapy were blood iron decrease, upper respiratory tract inflammation, nasopharyngitis, stomatitis, lymphocyte decrease, AST increase and ALT increase. Although most of these AEs were predominantly mild or moderate in severity; however, the severe liver injury was noted [76,77,112]. Additionally, it seems that efficacy and toxicity of iguratimod is dependent on genetic polymorphism [113].

AEs resulting from BTK inhibition can be followed in clinical trial of spebrutinib in RA patients [81]. It was well tolerated and most AEs like nausea, back pain, diarrhea, cough, and migraine were mild in severity. However, as preclinical toxicology studies in mice showed maturing spermatid degeneration, spebrutinib was taken so far only by female patients, which limits the potential drug usage [81].

## 3. The Role of tsDMARDs in Fighting COVID-19

Since 11 March 2020, WHO declared disease COVID-19, due to SARS-CoV-2 infection, as a pandemic [114]. In more severe cases, the disease is characterized by interstitial pneumonia with alveolar damage, which can lead to cytokine release syndrome (CRS) associated with massive production of pro-inflammatory cytokines and chemokines (IFN-α, IFN-β, IL-1β, IL-6, IL7, IL-8, IL-2, TNF, CXCL10, CCL2) within approximately 7 days or later of symptom onset [114,115]. The control of this cytokine storm is the major unmet need in COVID-19 treatment. SARS-CoV-2 enters targeted cells through receptor, angiotensin-converting enzyme 2 (ACE2)-mediated endocytosis [116]. Some regulators of this endocytosis belong to numb-associated kinase (NAK) family, such as AP2-associated protein kinase 1 (AAK1) and cyclin G-associated kinase (GAK) [117]. Inhibition of AAK1 would block the access of the virus into lung cells. However, to treat severe cases of COVID-19, when the host inflammatory response becomes a major cause of lung damage and subsequent mortality, there is a need to identify drugs combining anti-viral and anti-inflammatory properties [118].

The increasing knowledge about the SARS-CoV-2 infection is leading to consider some anti-rheumatic tsDMARDs as potential treatment options for the management of COVID-19. Indeed, the group of five JAKi, which are already approved for medical treatment, including tofacitinib, baricitinib, ruxolitinib, upadacitinib and fedratinib may be a good candidate. In particular, baricitinib has high affinity for AAK1 and ability to ameliorate chronic inflammation in interferonopathies [118]. Indeed, baricitinib inhibits NAK family, suggesting that baricitinib could not only repress the cytokine storm, but also block the virus entry into the host cells. In addition, it has been demonstrated that patients treated with baricitinib had reduced fever, breathlessness, cough, CRP levels and improvements in pulmonary function tests [119]. Importantly, JAK-STAT pathway blocking by baricitinib, results in impairment of IFN signal, which is one of the strongest innate immune responses preventing viral replication [120]. Therefore, introducing baricitinib at the late stage of disease, in order to control dangerous cytokine storm, but not to stop virus clearing by IFN, might be a reasonable proposal [115]. The risk of serious infections strongly depends on disease activity, supporting the importance of maintaining a good disease control in order to reduce infectious complications [121]. It has been demonstrated that Eli Lilly has started phase II/III/IV trial of baricitinib in COVID-19 as a mono therapy treatment and in combination with other drugs (clinical trials no. in Table 1). Many clinical trials have been also started with using ruxolitinib to combat COVID-19 (clinical trials no. in Table 1). The other JAKi, tofacitinib, is currently in a phase IIb randomized double blinded placebo controlled study for the treatment of moderate COVID-19 (clinical trials no. NCT04469114, NCT04415151). Importantly, p38 MAPK pathway also plays a crucial role in the release of pro-inflammatory cytokines and is activated by Angiotensin II. Angiotensin II is converted by ACE2, suggesting that blocking of p38 may impact multiple components of COVID-19. In vitro study has showed protective effects of p38 inhibition in a SARS-CoV [122]. Previous clinical study already demonstrated that losmapimod (p38 inhibitor) is involved in the reduction of pro-inflammatory biomarkers such as CRP and IL-6; therefore, the FDA approved phase III clinical study of losmapimod as a potential treatment for COVID-19 (LOSVID clinical trial). Losmapimod is already approved for facioscapulohumeral muscular dystrophy treatment [123]. Furthermore, recent data suggested a potential role of SYK-BTK inhibition in preventing thrombosis during COVID-19 [124]. Indeed, the authors speculated that SYK-BTK inhibition may block platelet-activating receptor (CLEC-2) and may reduce microvascular and venous thrombosis in COVID-19 patients. In addition, a mouse model study has demonstrated that fostamatinib (SYK inhibitor) can promote Mucin-1 removal from the surface of mucosal epithelial cells [125]. Increased expression of Mucin-1 is a biochemical marker predicting the development of acute lung injury. Furthermore, off-label clinical study demonstrated that treatment with the acalabrutinib (BTK inhibitor) resulted in decreased inflammation and improve outcomes in COVID-19 patients [126]. An alternate strategy for treating CRS in COVID-19 are also biologic agents including anti-IL-6Rα (tocilizumab and sarilumab), anti-IL-1 (anakinra), anti-CD20 (rituximab) and anti-GM-CSF (mavrilimumab, lenzilumab, gimsilumab); however, to evaluate the efficacy of these therapeutic modalities are still ongoing. 

Although there is increased infectious risk in RA due to the disease itself and immunosuppressive agents administration, recent observations showed that patients with chronic arthritis treated with bDMARDs or tsDMARDs do not seem to be more prone to life-threatening complications from SARS-CoV-2 than general population [127,128]. Guiding principles from EULAR and ACR suggested that RA patients should continue treatment during COVID-19 and to allocate adequate supplies of IL-1 and IL-6 and JAK antagonists for especially those RA patients, in whom even brief drug holidays would be expected to cause a flare of their disease [129,130,131,132]. Interestingly, recent clinical study revealed that the corticosteroid dexamethasone can reduce mortality in severe COVID-19 patients receiving oxygen therapy [133]. This suggests that dexamethasone, commonly used immunosuppressive agent for RA treatment, may improve the clinical outcome of hospitalized patients with severe COVID-19. 

## 4. Conclusions 

Despite extensive research and clinical efforts have been made in the past years, when the first tsDMARDs have been approved by FDA in 2012, the importance of development of more specific molecules is still needed in order to improve the quality of life for patients with active RA. 

This review critically analyze the benefits and disadvantages of latest clinical trials using newly discovered small molecule inhibitors with particular focuses on inhibition of JAKs/MAPKs/NF-κB/SYK-BTK signaling pathways. One advantage of tsDMARDs treatment is that these small molecules are orally administered and can offer wide protection against pro-inflammatory cytokines, in opposition to bDMARDs, which block specific extracellular molecules.

However, further studies are needed in order to monitor their risk-benefit ratio including increased risk of infection and thromboembolism. Indeed, recently, the EMA and FDA released warnings about the risk of blood clots in patients taking high dose of tofacitinib [99,100]. Therefore, the performance of blood tests or viral infection screening should be mandatory before the tsDMARDs administration.

Recent events of SARS-CoV-2 outbreak have demonstrated the potential use of small molecules inhibitors not only in treatment of RA but also in fighting COVID-19 via blocking the viral entry, preventing of hyperimmune activation and reducing cytokine storm. However, the administration of tsDMARDs should be carefully monitored by clinicians during each stage of COVID-19 progression. In particular, during early stage of COVID-19, small molecules treatment might be considered as beneficial in blocking viral entry, while during late stage of disease (>7 days), administration of tsDMARDs should be helpful in inhibition of live threatening cytokine storm production. Importantly, EULAR and ACR guidelines and recommendations demonstrate that there is no higher risk of COVID-19 susceptibility upon immunosuppressant treatment in RA patients.

Overall, this review provides an update regarding small molecule inhibitors as a novel anti-rheumatic drug and as a potential strategy for fighting COVID-19; however, it will require more study to validate the current results and to determine the right protocols.

## Figures and Tables

**Figure 1 cells-09-01876-f001:**
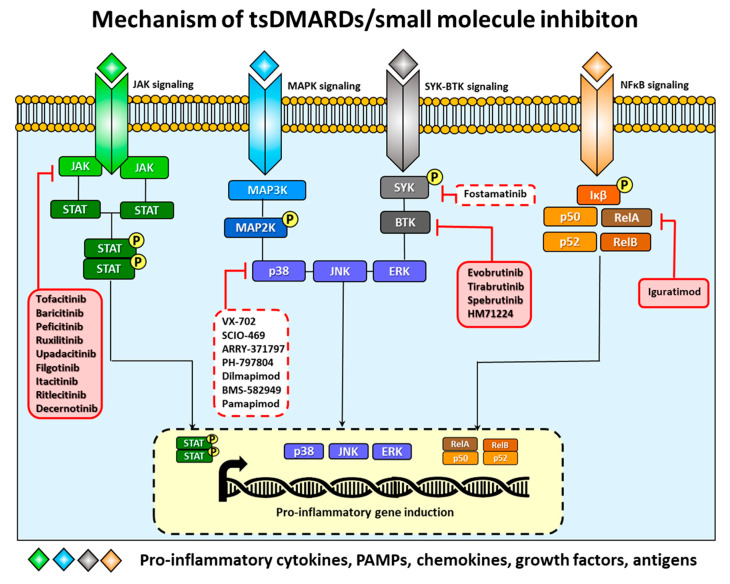
The mechanism of target synthetic DMARDs (tsDMARDs) inhibition in rheumatoid arthritis (RA). tsDMARDs, also known as small molecule inhibitors, are involved in blocking several pro-inflammatory pathways including JAK/MAPK/SYK-BTK/NF-κB signaling. tsDMARDs which are currently investigated in clinical trials or already approved for RA treatment are highlighted by the red rectangles, tsDMARDs which are terminated are highlighted by the white rectangles with dashed line.

**Table 1 cells-09-01876-t001:** Kinase and other small molecule inhibitors investigated in RA.

Target	Compound Name	Company	Current Development Phase in RA	Current Development in COVID-19
JAK1/JAK3	Tofacitinib	Pfizer	Approved by FDA and EMA	phase II(NCT04469114, NCT04415151)
JAK1/JAK2	Baricitinib	Eli Lillyand Company	Approved by FDA and EMA	phase II/III/IV (NCT04358614, NCT04421027, NCT04340232, NCT04346147, NCT04390464, NCT04320277, NCT04373044, NCT04321993, NCT04366206)
JAK 1,2,3TYK2	Peficitinib(ASP015K)	Astellas Pharma, Inc.	Approved in Japan	
JAK 1/2	Ruxolitinib(INCB018424)	Incyte Corporation	phase II (NCT00550043) completed	phase II/III(NCT04359290, NCT04359290, NCT04362137, NCT04348071, NCT04355793, NCT04377620, NCT04334044, NCT04331665, NCT04366232, NCT04374149, NCT04338958, NCT04348695)
JAK1	Upadacitinib(ABT 494)	AbbVie	Approved by FDA and EMA	preclinical studies
JAK1	Filgotinib(GLPG0634, GS6034)	Galapagos NV	phase III (NCT03025308) activephase II (NCT02065700) activephase II (NCT03926195) recruiting	
JAK 1	Itacitinib(INCB039110)	Incyte Corporation	phase II (NCT01626573) completed	
JAK3/TEC	Ritlecitinib(PF-06651600)	Pfizer	phase II (NCT02969044) completed	
JAK3	Decernotinib(VX-509)	Vertex Pharmaceuticals	phase II/III (NCT01830985) completed	
NF-κB	Iguratimod	Jiangsu Simcere Pharmaceutical Co., Ltd.	Approved in Japan and China	
BTK	Evobrutinib (M2951)	Merck/EMD Serono Research & Development Institute	phase IIa (NCT02784106) completed; phase IIb (NCT03233230) completed	
BTK	Tirabrutinib(GS-4059)	Gilead Sciences	phase I (NCT02626026) completed	
BTK	Spebrutinib (CC-292)	Celgene	phase II (NCT01975610) completed	
BTK	HM71224	Hanmi Pharmaceutical Co., Ltd.	phase I (NCT01765478) completed

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
