# Peer review of "Small Molecule Inhibitors in the Treatment of Rheumatoid Arthritis and Beyond: Latest Updates and Potential Strategy for Fighting COVID-19"

_cells, 2020, doi:10.3390/cells9081876_

Round 1

Reviewer 1 Report

This manuscript is a review of small molecule inhibitors that are currently in use or in development for the treatment of Rheumatoid Arthritis.  It also includes material concerning the usage of these treatments in  covid-19 infected patients. 

The review is thorough and well researched.  The extension into the use of these reagents for cover-19 is timely and important material for clinicians to have.

I would suggest, however, that the authors should clarify that the inhibition of the immune system is more appropriate in the later stages of Covid-19 and not in the early (first 7 days) stages of the disease.  There is concern that inhibiting the immune system early will deter the ability of the patients immune system to fight the virus since we have no current treatment.  To be fair, the authors do include one sentence to this effect, but more clarification would be helpful to the clinician readers.

Author Response

Thank you for this comment. We have addressed this comment. Pleases see the line 489-494 (page 11).

Reviewer 2 Report

This is a concise well-written and timely review and thromboembolism and COVID-19 aspects increase the novelty score.  It would clarify the presentation if subtitles for different inhibitor classes (JAK, MAPK...) were used, in addition  I have only minor, mainly wording suggestions:

  • page 4, line 152: aims to target selectively
  • page 4, line 158: targeting JAK1 and JAK3, and JAK2 to lesser extent
  • page 6, lines 279-281. At the moment the selectivity of JAK inhibitors is not at the level that inhibition of “selected cytokine” is possible. Sentence should be rephrased accordingly.
  • page 6, line 283: “the most feared” should be rephased to e.g. the most expected.
  • page 7, lines 311-313. Is increased risk of thromboembolism related also to ruxolitinib treatment in myelofibrosis patients? Or tofacitinib in treatment of psoriatic arthritis, ankylosing spondylitis or ulcerative colitis?
  • page 7, line 340. “All JAKi” is not correct. Covalent Pfizer JAK3 inhibitor does not inhibit JAK2. Could highly selective JAKinibs that do not inhibit JAK2 be a key to circumvent thromboembolism side-effect.
  • page 7, lines 346-347. IC50 data of upadacitinib for JAK1 vs JAK2 could be added.
  • page 9, line 399. Rephraze “five selective JAKi”. Three of these are first-generation non-selective inhibitors.
  • page 9, lines 407-408. The meaning of the sentence is unclear. Rephrase.
  • page 9, lines 434-435. “, however the RA patients should not stop taking these medications.” could be excluded for clarity.
  • page 10, line 459. High risk could be rephrased to increased risk (the risk for thromboembolism in general is not high).
  • page 10, lines 465-470. More suitable in Introduction than in Conclusions. Rephrase to include the concluding notes rather than introducing the review.
  • Table 1. For clarity, change the column heading “Current development phase in RA” and exclude RA from individual cells. A column for current development stage in COVID-19 could be added.
  OS  

Author Response

This is a concise well-written and timely review and thromboembolism and COVID-19 aspects increase the novelty score.  It would clarify the presentation if subtitles for different inhibitor classes (JAK, MAPK...) were used, in addition  I have only minor, mainly wording suggestions:

Thank you for the comment. We have divided the manuscript including the different inhibitor classes (JAK, MAPK, etc). Please see amended section 2.2.1 and 2.2.2. We did not divide the 2.2.3 chapter since the AEs of MAPKs, NF-kB and SYK-BTK inhibitors cover only a small fragment of this section.

  • page 4, line 152: aims to target selectively

We have addressed this comment. Pleases see the line 168 (page 4)

  • page 4, line 158: targeting JAK1 and JAK3, and JAK2 to lesser extent

We have addressed this comment. Pleases see the line 174 (page 4)

  • page 6, lines 279-281. At the moment the selectivity of JAK inhibitors is not at the level that inhibition of “selected cytokine” is possible. Sentence should be rephrased accordingly.

We have addressed this comment. Pleases see the line 299-301 (page 7)

  • page 6, line 283: “the most feared” should be rephased to e.g. the most expected.

We have addressed this comment. Pleases see the line 303-304 (page 7)

  • page 7, lines 311-313. Is increased risk of thromboembolism related also to ruxolitinib treatment in myelofibrosis patients? Or tofacitinib in treatment of psoriatic arthritis, ankylosing spondylitis or ulcerative colitis?

We have addressed this comment. Pleases see below.

We have analysed the recent FDA Adverse Events Reporting System (FAERS) Public Dashboard dated 30 June 2020 trying to answer your question.

Investigating “decitabine/ruxolitinib; ruxolitinib phosphate/ruxolitinib” treatment, among 240 cases with myelofibrosis there were: 2 cases of thrombosis (including 1 death), portal vein thrombosis and retinal vascular thrombosis.

Investigating “tofacitinib; tofacitinib citrate” in psoriatic arthropathy or juvenile psoriatic arthritis, we found total cases of 1441 AE reports in those diseases. Thrombosis (including pulmonary thrombosis, cerebral artery thrombosis, coronary artery thrombosis or deep vein thrombosis) was noted in 8 cases among them (7 serious cases and 1 death).

Investigating “tofacitinib; tofacitinib citrate” in ankylosing spondylitis, we found 145 cases of AE (including 58 serious cases and no deaths), however we could not find thrombosis cases using this website.

Investigating “tofacitinib; tofacitinib citrate” in ulcerative colitis, we found 1919 cases of AE (including 1027 serious cases and 26 deaths). Thrombosis (including pulmonary thrombosis, coronary artery thrombosis, retinal vein thrombosis, aortic thrombosis, venous thrombosis or deep vein thrombosis) was noted in 40 cases among them (39 serious cases and 1 death).

  • page 7, line 340. “All JAKi” is not correct. Covalent Pfizer JAK3 inhibitor does not inhibit JAK2. Could highly selective JAKinibs that do not inhibit JAK2 be a key to circumvent thromboembolism side-effect.

We have addressed this comment. Pleases see the line 360-367 (page 8)

  • page 7, lines 346-347. IC50 data of upadacitinib for JAK1 vs JAK2 could be added.

We have addressed this comment. Pleases see the line 372-373 (page 8)

  • page 9, line 399. Rephraze “five selective JAKi”. Three of these are first-generation non-selective inhibitors.

We have addressed this comment. Pleases see the line 425 (page 9)

  • page 9, lines 407-408. The meaning of the sentence is unclear. Rephrase.

We have removed this sentences.

  • page 9, lines 434-435. “, however the RA patients should not stop taking these medications.” could be excluded for clarity.

We have addressed this comment. Pleases see the line 462-465 (page 10).

  • page 10, line 459. High risk could be rephrased to increased risk (the risk for thromboembolism in general is not high).

We have addressed this comment. Pleases see the line 483 (page 10)

  • page 10, lines 465-470. More suitable in Introduction than in Conclusions. Rephrase to include the concluding notes rather than introducing the review.

We have addressed this comment. Pleases see the line 479-481 (page 10) and 489-494 (page 11)

  • Table 1. For clarity, change the column heading “Current development phase in RA” and exclude RA from individual cells. A column for current development stage in COVID-19 could be added.

We have addressed this comment. Pleases see amended Table 1.